# Health Systems, Aging, and Inequity: An Example from Chile

**DOI:** 10.3390/ijerph17186546

**Published:** 2020-09-09

**Authors:** Pablo Villalobos Dintrans

**Affiliations:** Programa Centro de Salud Pública, Facultad de Ciencias Médicas, Universidad de Santiago, 8320000 Santiago, Chile; pvillalobos.d@gmail.com

**Keywords:** Chile, health systems, aging

## Abstract

*Background:* Just like many other countries around the world, Chile is facing the challenges of demographic transition and population aging. Considering this context, the question of how prepared the health system is to deal with these challenges arises; *Methods:* A framework to assess the health system’s preparedness for aging was proposed, considering the health system’s goals and features and using an equity approach. Indicators related to the health system’s goals were calculated for the year 2015 using three nationally-representative sources: health status (suicide rate), financial protection (out-of-pocket and catastrophic expenditures), and responsiveness (satisfaction). Age ratios were used to compare the system’s response to different age groups; *Results:* Results for Chile revealed the existence of inequities, particularly when assessing the system in terms of its ability to improve health status and financial protection. These gaps increase with age, suggesting that the Chilean health system is not prepared to meet older people’s needs; and *Conclusions:* These results call for a reform in the health system, as well as the need for implementing a long-term care system in the country.

## 1. Introduction

Chile is experiencing a phase of accelerating aging. While the country’s share of older people (65 years and older) was 3.4% in 1950, this figure reached 12.2% in 2020. The number will double in the next 25 years, and the share of older people will exceed 30% by 2065. The same change, i.e., an increase in the share of older people from 10% to 20%, occurred in developed countries over a span of 70 to 100 years [1]. A second important feature of the process of demographic transition in the country is that the group that will increase the most in the coming years will be the “oldest among the older”, i.e., people over 80 years [2]. While the five-year average growth in the percentage of people 65+ for the next 80 years will be 6.8% in the country, the growth in the proportion of people 80+ will be 12.2% for the same period. Consequently, the share of people 80+ in the group of older people will rise from 11.6% in 2020 to 30% in 2040, and 49% by 2100 [1].

This process will impose several challenges to the country, particularly to its social security system: the increase in older people poses extra pressure on pensions, social policies, and the health system. The health system needs to adapt to these changes including, among other things, changes in the burden of disease and an increase in other conditions, such as long-term care needs. In fact, the World Health Organization has called to rethink health systems in this context of population aging, proposing among other things, the adaptation of health systems to respond to older people’s needs and the implementation of long-term care systems [3,4]. Several studies on Chile have identified population aging as the main challenge, highlighting the country’s lack of preparedness to deal with these changes [5,6,7]. The Chilean health system has also been assessed in terms of its ability to meet its goals. However, most studies focus either on one dimension of the system (e.g., access, financing protection) instead of a holistic assessment, or leave out a focus on older people as a population with specific health needs [8,9,10]. This article intends to fill this gap by proposing a comprehensive age-inequity approach.

Considering this scenario, asking whether the Chilean health system is prepared to deal with these challenges is an important question. The aim of the article is to propose a generic framework to assess the health system’s preparedness to deal with the challenges of an aging population and use it to test the Chilean health system’s response to older people’s health needs.

## 2. Materials and Methods

### 2.1. A Framework to Assess the Health System Preparedness for Aging

Selecting indicators to assess the health system performance in the context of increasing aging is not an easy task. In order to structure the process, a generic framework based on three components was proposed. These components were health system goals, benchmark strategy, and health system features.

#### 2.1.1. Health System Goals

In the first place, indicators need to be related to the goals of the health system. Based on different health system models [11,12,13,14], these goals can be broadly defined in terms of:i.Improving health statusii.Providing financial protectioniii.Meeting people’s expectations

These goals have usually been evaluated using indicators such as life expectancy, mortality, morbidity, disability, quality-adjusted life years, and self-reported health status, among others for health status [15,16,17,18]; other indicators include the share of out-of-pocket expenditures (OOPE), the presence of catastrophic health expenditures for financial protection [19,20], and some measure of people’s satisfaction with the health system as an assessment of the system’s responsiveness [21]. However, as discussed below, since several of these indicators are still helpful to assess the performance of the health system in regard to older people, many adjustments are required.

#### 2.1.2. Benchmark Strategy

A second step in selecting relevant indicators is choosing a benchmark; this allows emitting a judgment on the indicator’s value. Different dimensions of performance might require different benchmarks. In order to simplify the discussion, these have been classified into three categories:i.Evolution (time indicators)ii.Overall performance (cross-section between indicators)iii.Inequality (cross-section within indicators)

In the first case, the same indicator is compared to itself over time. In the second case, indicators of the health system—in this case, indicators that measure the health system performance with respect to older people—are compared to the same indicator in a different country/health system. Finally, indicators can also measure between different groups within a country/health system in a given period of time.

Although the first and second strategies are interesting, the focus of this article is on inequalities within the health system. Comparing between countries can be useful to set potential outcomes (aspirational goals), but has at least two important limitations: first, comparability between countries can be difficult due to methodological differences in the calculation of indicators in different countries; second, inter-country comparisons need to take into account the socioeconomic, epidemiological, and demographic differences between countries, particularly when the focus is on comparing preparedness to respond to the challenges of aging. The evolution of an indicator presents similar problems; as time goes by, the likelihood of methodological and context changes increases. Additionally, some indicators might need extended periods of time to capture changes. Finally, data availability and quality could also determine the mix of indicators and the way in which they should be interpreted.

When using inequality as benchmark strategy, it is necessary to assume that under an ideal health system, the three goals described above should be met regardless of the population’s characteristics, in terms of demographics (e.g., using age or gender to classify groups) or socioeconomic conditions (e.g., using income or education), for example. Consequently, the strategy is based on the idea that indicators should not be different for different groups (in this case elderly versus non-elderly populations).

The main challenge, in this case, is finding indicators that evidence not only the existence of disparities between elderly and non-elderly populations, as can be usually found in the literature [22,23], but indicators that reflect the presence of inequities related to the goals of the health system. Following the work of Margaret Whitehead [24,25], health inequalities can be considered inequities if they are (i) avoidable and unnecessary and; (ii) unfair and unjust. In this regard, disparities can be considered unjust (i) if they are socially produced, i.e., there are socially-controllable factors that explain those health disparities; and (ii) even assuming the existence of “natural” (e.g., genetic) factors that justify the differences, the society does not deal with them, even if they have the capacity to do so [26].

#### 2.1.3. Health System Features

Finally, using these two pieces (health system frameworks to define goals, and health equity frameworks to guide the selection of indicators), it is possible to propose a set of indicators to assess health system preparedness for aging in terms of equity. However, it is also required to take into account some features of the health system that could confound the interpretation of the results.

The framework proposed to evaluate the health system’s preparedness for population aging is depicted in Figure 1. It includes three sets of indicators related to the three goals of the health system. Each indicator can be evaluated using three benchmark selection strategies; as noted before, this article will focus on the presence of inequalities within the system. Finally, if necessary, results should also consider idiosyncratic features of the health system that could confound the results, such as the way in which healthcare services are provided, the presence of insurance schemes, and the share of health coverage.

### 2.2. Data Sources and Indicators

#### 2.2.1. Data Sources

The proposed framework was operationalized using data on the Chilean health system. This process required selecting specific indicators for the scheme proposed in Figure 1 and computing their values.

Data for calculating the indicators came from different sources and institutions. Data on deaths by cause is annually published by the Chilean Ministry of Health [27]. Data on health expenditure, used to calculate the health financial protection indicators, was obtained from the National Survey on Household Budgets by the Chilean National Institute of Statistics, carried out every five years to collect data on income and expenditures of Chilean families [28]. Finally, information on people’s satisfaction with the health system came from an annual survey carried out by the Chilean Superintendence of Health to assess the performance of the system [29]. To ensure comparability of the results, all indicators were calculated for the year 2015, the most recent information available for the three sources. Data has national representativeness.

#### 2.2.2. Indicators for the Assessment

The challenge of identifying a set of indicators becomes evident when thinking about the first goal of the health system: improving health status. Traditional indicators are commonly used when assessing health inequalities between different groups—for example by income or race—since differences are directly interpreted as inequities. However, this is not the case when comparing populations using age as the criterion for clustering people. In this circumstance, using indicators based on mortality/morbidity will reveal expected differences that cannot be attributable to the health system’s performance. As stated before, in this situation, the indicator should, under an ideal health system, be equal between age groups, i.e., the expected ratio for the elderly and non-elderly indicators should be equal to one. In this case, the indicator used was suicide mortality rate in 2015—measured as the number of deaths by suicide for each age group over the total population of that group in the year 2015, multiplied by 1000—from the Ministry of Health’s Department of Statistics and Health Information [27].

When measuring preparedness in terms of financial protection, the problem detected for the health status indicators, i.e., the challenge of disentangling inequalities from inequities, does not appear; although health expenditures could differ between populations of different ages, reducing financial risk regardless of age, sex, or other conditions is exactly the role of the health system, since these differences are, in fact, avoidable. Consequently, disparities in financial risks should be considered as unfair. In this case, two measures were used: OOPE as the share of total household expenditure, and the percentage of households that incur catastrophic expenditure (using a 30% threshold). Data come from the 2015 Family Budget Survey, conducted by the Chilean National Institute of Statistics [28]. Since data on health expenditure come from a household survey, the comparison groups are not elderly versus non-elderly populations, but households with and without older people. The share OOPE/expenditure was calculated using the total OOP health expenditure and total expenditure reported; a household was considered as incurring catastrophic expenditure if the ratio of total health expenditure/ability to pay was larger than 30%, with ability to pay defined as the total household expenditure minus the expenditure on food and (non-alcoholic) beverages following the methodology proposed by [30].

People’s satisfaction with the health system could be measured in different ways [21,31]; however, any indicator should explicitly incorporate a measure of how satisfied people are with the health system. In this case, the indicator used was general (self-reported) satisfaction with the health system from the Users Opinion Study/Health System, collected by the Superintendence of Health in 2015 [29]. The scale ranges from 1 (worst) to 7 (best).

Finally, as explained in the previous section, when evaluating differences in the health system’s goals using age as the criterion, the assessment has to incorporate features of the system that could confound the interpretation of the results. One salient characteristic of the Chilean system is the presence of selection in the health insurance market, which leads to a high degree of segmentation; the public insurer—National Health Fund (*Fondo Nacional de Salud*, FONASA)—ends up covering high-risk (in terms of sex and age) and low-income populations [32,33]. Given that age is highly correlated with health insurance, using age to compare indicators can mislead the interpretation since these could be explained not only by the difference between elderly and non-elderly populations, but due to differences between insurance schemes. In order to overcome this problem, results were disaggregated by type of insurance: public insurance: FONASA and private insurance (*Instituciones de Salud Previsional*, ISAPRE).

## 3. Results

A first set of results is presented in Table 1. The first row shows the value of the indicator for Chile, and the following rows show the ratio for elderly and non-elderly populations, using 65 years as the threshold. The group 65+ includes all people with 65 years and older.

The first indicator related to the goal of improving population health shows that suicide rates were larger in the elderly group, reflecting a difference in the way in which the system contributes to the health status of both groups of the population. Unfortunately, the ratios cannot be calculated by insurance scheme since the Ministry of Health does not identify this information.

When analyzing the indicators for financial protection, results exhibited the presence of important gaps between elderly and non-elderly populations, as well as other interesting patterns. First, the gap widened when comparing FONASA and ISAPRE, meaning that, in relative terms, financial protection for older people is worst in the private scheme (actually this is also true in absolute terms). Second, differences were larger when using catastrophic expenditure as a measure for financial protection; in this case, the disparities between the public and private insurance also held. As a result, the share of households incurring catastrophic health expenditure in households with older people was almost four times the same share in those households with no older people. These results confirm previous results showing the ineffectiveness of the private health system to provide financial protection to vulnerable groups [33,34].

In terms of satisfaction with the health system, differences were smaller (ratios were close to one), with satisfaction being relatively higher for the elderly in FONASA (compared to ISAPRE). For this indicator, data showed that satisfaction was higher for elders (except for those in ISAPRE). Results can be explained by the existence of self-reported bias; self-reported satisfaction was a complex measure that comprised expectations, access, and quality of services, among other factors. As found in studies using self-reported health status, differences can be observed between subjective (self-reported) and objective (mortality, morbidity) measures, with people with better average health indicators reporting having worst health and vice versa [35,36]. Despite the reasons, the indicator directly captured people’s satisfaction with the health system, regardless of their access and quality of service.

Finally, an extra set of results was presented using different definitions for the age groups. The rationale for this is two-fold. First, the threshold for what is considered an "older person" is not clearly defined [37]; when looking at studies and regulations, two age thresholds—60 and 65 years old—are commonly used, although 65 years is usually accepted as the definition in developed countries [38,39]. Second, a particular feature of the aging process in countries like Chile is not just that people are getting older, but that the oldest among the older are rising very fast; in the next few years, the share of the population over 65 years will grow 4% per year, with the share of people over 65 years going from 11% of the population to 20% in 20 years (by 2038). However, population over 80 years will grow even faster (4.5% average for the next 20 years); by 2100, 30% of the population will be over 65 years, and half of them will be older than 80 years old [2]. Table 2 shows the results for different groups of older people.

As before, indicators showed a bias against the group of older people, except when looking at the satisfaction with the health system. The main result from the table was that these gaps increase with age for each indicator. This is a very important result, since it can be interpreted as a failure of the health system to meet the needs of older adults in the country, with these inequities increasing with age.

## 4. Discussion

The aim of the article was to assess the Chilean health system’s preparedness to deal with the challenges of an aging population. It presented a framework to select indicators and understand their interpretation. The exercise showed that measuring health systems’ preparedness has several issues that need to be taken into account, particularly the features of the health system and the distinction between inequalities and inequities.

Although there are other frameworks proposed to assess health systems, they usually look at overall performance, missing some important details regarding the allocation of resources and inequalities between different population groups within a country [15,40]. On the other hand, there is a vast literature on the need to adapt health systems to the challenges of population aging [41,42,43,44]. However, these articles usually advocate for a reform rather than presenting the rationale for changing; they use the system’s perspective instead of a beneficiaries’ approach [45]. This article provides a complementary point of view, in which the rationale for a reform of the health system is based not just on the need for the system’s sustainability, but also on improving the system’s performance regarding older people.

The application of this framework to the Chilean health system reveals several interesting points. First, there are important inequities when comparing the health system’s performance for different age groups; these differences hold when taking into account the selection in the health insurance market. Second, related to this point, the private insurance scheme increases these differences, particularly in terms of financial protection. Third, gaps in results between elderly and non-elderly spread as age increases, evidencing the system’s lack of preparedness to deal with the needs of older people.

The study presented several limitations that should be considered when interpreting the results. First, although a general framework to select indicators and dimensions to assess them was presented, the evaluation was based on a very limited set of variables; although any assessment of the health system performance should consider indicators for its three goals, others indicators could also be used, depending on the context of the evaluation, data availability and quality, and each health system’s priorities. The exercise for Chile could serve as an example for other countries trying to assess their health system’s preparedness for population aging. In a similar vein, the number of indicators was limited. This decision was taken as a way to provide an example for Chile, but also to consider general indicators that could be easily measured in a different setting; the choice reflects an inherent trade-off between simplicity and comprehensiveness that is not easy to solve. Third, the list of indicators was also limited by using inequity as the benchmark strategy; these and other indicators could be added to the analysis. This is particularly important for the indicators measuring health status. For example, using the “Overall performance” strategy, Chile performs worse than the Organization for Economic Co-operation and Development (OECD) average in terms of life expectancy at age 65 (18.5 versus 19.5 years) [46], but better in terms of disability-adjusted life years (74,471.2 versus 72,632.4 per 100,000) [47].

## 5. Conclusions

The exercise proposed a way to assess how prepared a health system is to face the challenges of an aging society. Despite its limitations, it gave important information, not to evaluate the system’s general performance, but to identify trends and areas for improvement.

Like Chile, many countries around the world will face the challenges of an aging population in the coming years. The article showed that adapting health systems is not only important to ensure sustainability, but also to adjust to form systems that, although being designed to deal with different problems, are not responding to older people’s health needs. Combining both issues—an increasing share of older population and an underperformance of the system regarding older people—implies an increase of age-related health inequities around the world. The proposed framework allows countries—mutatis mutandis—to identify current inequities that are usually ignored. This adds a different perspective to the need of adapting health systems for population aging, as it is not just something countries should do in preparation for future challenges, but something that countries should do today to address institutionalized health inequities.

Results from the Chilean health system demonstrate the need to rethink the health system to better respond to the country’s current and future needs. It also highlights the relevance of incorporating long-term care systems as a complement of the health system to provide integral and coordinated solutions to the elderly population. Although older age and long-term care needs are synonymous, the two concepts are closely related, and the projected increase in the share of an older population in Chile is expected to increase the demand for long-term care services in the country [6]. In this scenario, a long-term care system is important, not just as a coordinated way to provide services to people with long-term care needs, but also to relieve pressure from the health system [7]. This complementarity also contributes to the need to improve the efficiency and performance of the health system, particularly for older people.

We encourage its application in other contexts and hope it will contribute to promoting the debate about the role of the health system in a changing world.

## Figures and Tables

**Figure 1 ijerph-17-06546-f001:**
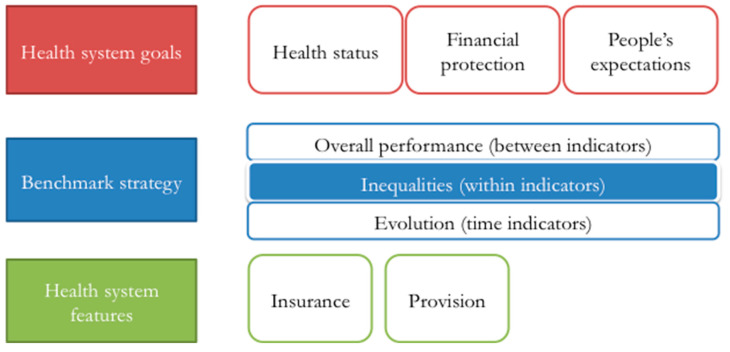
Indicators to assess health system performance for the elderly population.

**Table 1 ijerph-17-06546-t001:** Age ratio (65+/65−) indicators by insurance scheme.

Health System Indicator	Health Status	Financial Protection		People’s Expectations
Indicator	Suicide Mortality Rate (per 1000 People)	OOPE/Expenditure	Catastrophic Expenditure (% Households)	Satisfaction with the Health System (Scale from 1 to 7)
National level	0.10	4.19%	2.13%	4.96
Ratio 65+/65− (total)	1.32	1.49	2.68	1.07
Ratio 65+/65− (FONASA)	No data	1.53	2.54	1.09
Ratio 65+/65− (ISAPRE)	No data	1.74	3.81	0.99

**Table 2 ijerph-17-06546-t002:** Age ratio indicators for different age groups.

Health System Indicator	Health Status	Financial Protection		People’s Expectations
Indicator	Suicide Mortality Rate (per 1000 People)	OOPE/Expenditure	Catastrophic Expenditure (% Households)	Satisfaction with the Health System (Scale from 1 to 7)
Ratio 60+/60− (total)	1.31	1.44	2.48	1.01
Ratio 65+/65− (total)	1.32	1.49	2.68	1.07
Ratio 80+/80− (total)	1.44	1.69	3.90	1.15

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
