# Peer review of "Health Systems, Aging, and Inequity: An Example from Chile"

_ijerph, 2020, doi:10.3390/ijerph17186546_

Round 1
Reviewer 1 Report
Thank you for inviting me to review this manuscript Health systems, aging, and inequity: an example from 2 Chile.
The manuscript is easy to read and all data, and conclusions will present huge interest with impact on clinical practice especially in low and middle income countries.
I have only one proposal:
To extend more discussions because in my opinion is extremely short. Here you must compare your data with another countries from the region and same income, also it will be illustrative comparison and with some high income countries.
Author Response
Reviewer 1
Thank you for inviting me to review this manuscript Health systems, aging, and inequity: an example from 2 Chile.
The manuscript is easy to read and all data, and conclusions will present huge interest with impact on clinical practice especially in low and middle income countries.
I have only one proposal:
To extend more discussions because in my opinion is extremely short. Here you must compare your data with another countries from the region and same income, also it will be illustrative comparison and with some high income countries.
R: I included a paragraph in the discussion with several references regarding the efforts for measuring health systems’ performance, and advocating for adapting health systems for older people. The article tries to put together these two pieces, which are usually treated independently. This data is also presented at the end of the Discussion section.
Reviewer 2 Report
This manuscript entitled: Health systems, aging, and inequity: an example from 2 Chile - the aim of the study described in this manuscript is to The aim of the article is to propose a generic framework assess the health system’s preparedness to deal with the challenges of an aging population and use it to test the Chilean health system readiness to deal with this challenge.
I believe that this is a worthwhile important study that should be published. It is clearly written, well organized and interesting. Social inequality and access to health care is an important global concern - I would find more detailed information regarding the state of affairs in Chile helpful and important for this paper (overview statistics).
I do have a few comments that I would urge the authors to consider when when they prepare the final manuscript for publication.
The introduction needs to be expanded and it should include statistics of the aging population in Chile.
The authors state that the suicide rate is higher for those over 65 than under but provide not introduction or detail into this important finding.
This paper addresses satisfaction with access to the Chilean health care system - making assumptions about the preparedness of the system. It is an important topic and the paper, in my opinion, should be published, however the authors should flush out the details of the circumstances in Chile - after reading this paper I still do not know what percentage of the population is over 65, what percentage live in poverty, and so forth. An overview would contextualize the information.
Suggestions:
Provide actual data on the aging population in Chile.
Author Response
Reviewer 2
This manuscript entitled: Health systems, aging, and inequity: an example from 2 Chile - the aim of the study described in this manuscript is to The aim of the article is to propose a generic framework assess the health system’s preparedness to deal with the challenges of an aging population and use it to test the Chilean health system readiness to deal with this challenge.
I believe that this is a worthwhile important study that should be published. It is clearly written, well organized and interesting. Social inequality and access to health care is an important global concern - I would find more detailed information regarding the state of affairs in Chile helpful and important for this paper (overview statistics).
I do have a few comments that I would urge the authors to consider when when they prepare the final manuscript for publication.
The introduction needs to be expanded and it should include statistics of the aging population in Chile.
R: The information was added in the Introduction. Two paragraphs with demographic statistics describing the aging process in the country were added.
The authors state that the suicide rate is higher for those over 65 than under but provide not introduction or detail into this important finding.
R: The discussion on the selection of the variables explains why suicide rates can be an interesting indicator to assess the health system’s performance. The fact that suicide rate is higher for people 65+ is one of the main results and, consequently, is presented in the “Results” section (Tables 1 and 2).
This paper addresses satisfaction with access to the Chilean health care system - making assumptions about the preparedness of the system. It is an important topic and the paper, in my opinion, should be published, however the authors should flush out the details of the circumstances in Chile - after reading this paper I still do not know what percentage of the population is over 65, what percentage live in poverty, and so forth. An overview would contextualize the information.
R: The information was added in the Introduction. Two paragraphs with demographic statistics describing the aging process in the country were added.
Suggestions:
Provide actual data on the aging population in Chile.
R: The information was added in the Introduction. Two paragraphs with demographic statistics describing the aging process in the country were added.
Reviewer 3 Report
The reviewer wants to congratulate the author for his work, very interesting and appropriate for these days. However, some minor revisions must be made for publication:
- Eliminate the numbers from the abstract, it is advisable to write the paragraph continuously and without interruptions.
- The introduction should be expanded with some more reference to similar studies carried out in Latin America.
- Has the adjustment in the contribution and the pension system been taken into account?
- The conclusions must be expanded
Author Response
Reviewer 3
The reviewer wants to congratulate the author for his work, very interesting and appropriate for these days. However, some minor revisions must be made for publication:
- Eliminate the numbers from the abstract, it is advisable to write the paragraph continuously and without interruptions.
R: Agree. I used the journal’s template for the abstract. I’m OK with removing the numbers if they have no problem.
- The introduction should be expanded with some more reference to similar studies carried out in Latin America.
R: As far as I know, there are no similar studies published in Latin America. However, I included a paragraph in the discussion with several references regarding the efforts for measuring health systems’ performance, and advocating for adapting health systems for older people. The article tries to put together these two pieces, which are usually treated independently.
- Has the adjustment in the contribution and the pension system been taken into account?
R: As described in the methods sections, information comes from three different sources, and data corresponds to the year 2015. Unfortunately, this does now allow assessing recent changes in older people’s policies. However, the article is still useful to illustrate how health systems can be evaluated in terms of the readiness to meet older people's needs.
- The conclusions must be expanded
R: Conclusions were expanded with two new paragraphs. The first to highlight the equity dimension of the discussion, and the need to adapt health systems as a way to meet the actual needs of the population. The second paragraph extends the discussion on long-term care systems, as an important complement and help to improve the health system’s performance.
Round 2
Reviewer 1 Report
Accept in present form